# 4D Gaussian Splatting in the Wild with Uncertainty-Aware Regularization

**Mijeong Kim**[1]
mijeong.kim@snu.ac.kr

**Jongwoo Lim**[2, 3]
jongwoo.lim@snu.ac.kr

**Bohyung Han**[1,3]
bhhan@snu.ac.kr

[1]ECE, [2]ME, and [3]IPAI, Seoul National University, South Korea

## Abstract

Novel view synthesis of dynamic scenes is becoming important in various applications, including augmented and virtual reality. We propose a novel 4D Gaussian Splatting (4DGS) algorithm for dynamic scenes from casually recorded monocular videos. To overcome the overfitting problem of existing work for these real-world videos, we introduce an uncertainty-aware regularization that identifies uncertain regions with few observations and selectively imposes additional priors based on diffusion models and depth smoothness on such regions. This approach improves both the performance of novel view synthesis and the quality of training image reconstruction. We also identify the initialization problem of 4DGS in fast-moving dynamic regions, where the Structure from Motion (SfM) algorithm fails to provide reliable 3D landmarks. To initialize Gaussian primitives in such regions, we present a dynamic region densification method using the estimated depth maps and scene flow. Our experiments show that the proposed method improves the performance of 4DGS reconstruction from a video captured by a handheld monocular camera and also exhibits promising results in few-shot static scene reconstruction.

## 1 Introduction

Dynamic novel View Synthesis (DVS) aims to reconstruct dynamic scenes from captured videos and generate photorealistic frames for an arbitrary new combination of a viewpoint and a time step. This task has emerged as a vital research area in the 3D vision community with rapid advancements in augmented reality and virtual reality. Early DVS research primarily relied on neural radiance fields [29, 69, 10, 13, 11, 38, 39, 41, 5, 12, 50]. In contrast, more recent methods [61, 18, 31] extend 3D Gaussian Splatting [23] to account for the additional time dimension in dynamic scenes, and these techniques are referred to as 4D Gaussian Splatting.

Despite the recent success of 4D Gaussian Splatting models [61, 18, 31, 68], their applicability remains largely limited to controlled and purpose-built environments. Most existing models are developed and tested with multi-view video setups [29, 41]. While there are several methods tackling monocular video settings, these setups are still controlled and fall short of in-the-wild scenarios. For instance, [38, 69] maintain multi-view characteristics, where the camera captures a broad arc around a slow-moving object. Also, HyperNeRF [39] relies on unrealistic train-test splits, with both sets sampled from the same video trajectory, which renders the task closer to video interpolation than genuine novel view synthesis. In this paper, we focus for the first time on more natural, real-world monocular videos [14], where a single handheld camera moves around fast-moving objects.

In casually recorded monocular videos, which often lack sufficient multi-view information, 4D Gaussian Splatting algorithms tend to overfit the training frames in real-world scenarios. To address overfitting, recent regularization techniques [26, 7, 58, 67, 25, 36, 20] can be applied to provide

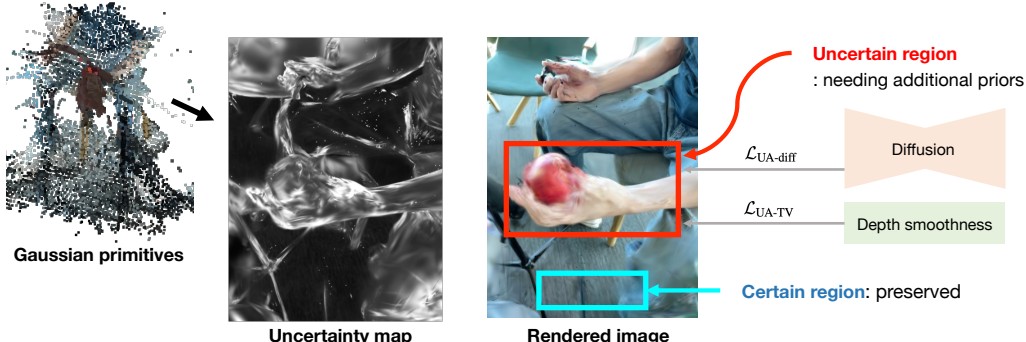

**Gaussian primitives**    **Uncertainty map**    **Rendered image**

Figure 1: Concept of uncertainty-aware regularization. Existing models often use regularization techniques to introduce additional priors for unseen views, aiming to enhance novel view synthesis performance. However, these methods tend to over-regularize accurately reconstructed pixels, which degrades the reconstruction quality of training images. To address this issue, our uncertainty-aware regularization selectively focuses on uncertain regions in unseen views, preserving the quality of well-reconstructed pixels with low uncertainty.

additional priors for unseen views. However, these regularization techniques often involve a balancing issue: while they effectively improve novel view synthesis performance during testing, they inherently sacrifice the reconstruction accuracy of training images. Since both the reconstruction accuracy and the novel view synthesis quality are equally important in our target task, the trade-off caused by the naïve application of the regularization techniques is not desirable.

In this paper, we address this balancing issue with a simple yet effective solution: uncertainty-aware regularization. First, we quantify the uncertainty of each Gaussian primitive based on its contribution to rendering for training images. Then, a 2D uncertainty map is constructed for unseen views using an $\alpha$-blending method. Regularization is selectively applied to uncertain regions, guided by the diffusion and depth smoothness priors, while low-uncertainty regions, where training data already provide sufficient reconstruction detail, are left unregularized, as illustrated in Figure 1. This approach results in a better balance between training and test performance, achieving good performance.

In real-world scenarios involving fast motions, especially in casually recorded videos, 4D Gaussian Splatting additionally faces considerable challenges with initialization. The algorithms based on Gaussian Splatting initialize Gaussian primitives using point clouds obtained by Structure from Motion (SfM) [47]. However, SfM struggles to reconstruct dynamic regions, particularly those with fast motion, often treating them as noise and leaving these areas without initialized primitives. Such an incomplete initialization disrupts training, causing primitives in static regions to be repeatedly cloned and split in an attempt to fill the dynamic areas. This can lead to an excessive number of primitives and, in some cases, out-of-memory issues. To address this limitation, we propose a dynamic region densification technique that initializes additional Gaussian primitives in dynamic regions.

We address the challenging problem of 4D reconstruction from an in-the-wild monocular video recorded casually with a handheld camera—a scenario that has been rarely explored. The main contributions of this paper are summarized as follows:

- We propose an uncertainty quantification technique based on contribution to training image rendering and introduce adaptive regularization techniques based on the uncertainty map, which balances between novel view synthesis performance and training image reconstruction quality.
- We address the issue of incomplete initialization in dynamic regions, emphasizing the importance of proper initialization in the training process of 4D Gaussian Splatting.
- We demonstrate the effectiveness of our algorithm on casually recorded monocular videos, showing improvements over baselines. Additionally, we validate the applicability of our method in few-shot static scene reconstruction.

The rest of this paper is organized as follows. Section 2 reviews related work and Section 3 discusses the basic concepts of 4D Gaussian splatting, which builds upon 3D Gaussian Splatting by integrating

deformation strategies. The details of our approach are described in Section 4, followed by the presentation of experimental results in Section 5. Finally, we conclude this paper in Section 6.

## 2 Related Work

### 2.1 Dynamic Novel View Synthesis

In recent years, significant advances have been made in novel view synthesis [34, 6, 15, 60, 35, 23]. Initially focused on static scenes, novel view synthesis has shifted towards dynamic scenes through the integration of motion modeling, now referred to as Dynamic novel View Synthesis (DVS). Early approaches [13, 10, 38, 5, 12, 50, 11] are largely driven by Neural Radiance Fields (NeRF). Some studies [13, 10] capture dynamics implicitly through temporal inputs or latent codes, and other approaches [38, 5, 12, 50, 11] focus on training the canonical NeRF and its deformation fields.

The introduction of 3D Gaussian Splatting (3DGS) [23] has marked a paradigm shift in novel view synthesis, leading to the development of 4D Gaussian Splatting (4DGS) [68, 18, 61, 31] for DVS. These 4DGS methods deform canonical 3D Gaussian primitives over time using additional deformation networks, which can be based on MLPs, learnable control points, Hexplane [5, 61], or polynomial functions. While these models excel at reconstructing dynamic scenes in controlled environments [38, 69, 29, 41, 39], they face significant challenges when applied to casually recorded monocular videos, posing substantial hurdles for real-world applications.

### 2.2 Regularization Techniques in Sparse Reconstruction

Casually recorded monocular videos typically provide limited multi-view information, as they are typically captured with a single handheld camera that only exhibits gentle motion. Consequently, reconstructing dynamic scenes from these videos is often regarded as a form of sparse reconstruction due to the lack of multi-view data.

In the context of sparse reconstruction, various regularization techniques have been proposed to mitigate overfitting on limited training images [36, 25, 20, 48, 22, 4, 49, 53, 24, 63, 43, 8, 59, 55, 16, 65, 62, 2, 17, 16]. These approaches generally involve rendering images or depth maps for unseen views to provide additional priors. For instance, some methods leverage depth priors based on estimated depths for novel views [43, 8, 59, 55, 16, 72], while others incorporate color smoothness constraints to enhance these views [25, 36]. Building on the success of diffusion models [44], recent approach [62] starts to incorporate diffusion priors to produce more realistic images of novel views. While these methods effectively enhance novel view synthesis performance at test time, they inherently compromise the quality of training image reconstructions. Since both the reconstruction accuracy and the novel view synthesis quality are equality important in our target task, the trade-off caused by the naïve application of the regularization techniques is not desirable.

### 2.3 Uncertainty Quantifications in Novel View Synthesis

Uncertainty estimation in novel view synthesis has primarily been explored with Neural Radiance Fields (NeRF) [29]. Pioneering approaches [37, 51, 52] re-parameterize MLP networks in NeRF using Bayesian models to compute the uncertainty of network predictions. Inspired by InfoNeRF [25], which considers entropy along rays for few-shot NeRFs, some studies [70, 66, 27] quantify uncertainty using the entropy of density along rays from a novel view. Additionally, Density-aware NeRF Ensembles [56] measures uncertainty by examining the variance in RGB images produced by an ensemble of models.

In contrast, uncertainty quantification in Gaussian Splatting [23] remains largely underexplored, with only a few works addressing this issue. Savant *et al.* [46] incorporate variational inference into the rendering pipeline, but this approach increases learnable parameters. Similarly, FisherRF [21] quantifies the uncertainty of Gaussian primitives by aggregating the diagonal of the Hessian matrix of the log-likelihood function; however, it is not straightforward to obtain a scalar value of uncertainty from the Hessian matrix for each Gaussian primitive. Our approach, on the other hand, directly quantifies the observed information of each Gaussian primitive by aggregating their contributions to the reconstruction of training images.

Most existing works that utilize estimated uncertainty primarily focus on quantifying model predictions after training [46] or on active learning for next-view selection [21]. In contrast, our approach leverages estimated uncertainty for adaptive regularization during training, specifically targeting Gaussian Splatting in sparse reconstruction.

# 3 Preliminary: 4D Gaussian Splatting

This section briefly overviews 3D Gaussian splatting (3DGS) [23] and explains the deformation modeling in 4D Gaussian Splatting (4DGS) [68, 18, 61, 31] for dynamic scenes.

## 3.1 3D Gaussian Splatting

**Gaussian primitive**  3D Gaussian splatting has demonstrated real-time, state-of-the-art rendering quality on static scenes. It uses an explicit 3D scene representation consisting of a set of 3D Gaussian ellipsoids, denoted by $\Gamma = \{\gamma_1, ..., \gamma_K\}$. Each Gaussian primitive, $\gamma_k$, is based on an unnormalized 3D Gaussian kernel, $\mathcal{G}_k(\boldsymbol{x})$, parameterized by $\boldsymbol{\mu}_k$ and $\boldsymbol{\Sigma}_k$ as follows:

$$\mathcal{G}_k(\boldsymbol{x}; \boldsymbol{\mu}_k, \boldsymbol{\Sigma}_k) \coloneqq \exp\left(-\frac{1}{2}(\boldsymbol{x} - \boldsymbol{\mu}_k)^\top \boldsymbol{\Sigma}_k^{-1}(\boldsymbol{x} - \boldsymbol{\mu}_k)\right), \tag{1}$$

where $\boldsymbol{\mu}_k \in \mathbb{R}^3$ is the center position, $\boldsymbol{\Sigma}_k \in \mathbb{R}^{3\times3}$ is an anisotropic covariance matrix, and $\boldsymbol{x} \in \mathbb{R}^3$ is an arbitrary location in 3D space. The covariance matrix $\boldsymbol{\Sigma}_k$ is valid only when positive semi-definite, which is challenging to enforce during optimization. To ensure this condition, we learn $\boldsymbol{\Sigma}_k$ by decomposing it into two learnable components, a rotation matrix $\boldsymbol{R_k}$ and a scaling matrix $\boldsymbol{S_k}$ as follows:

$$\boldsymbol{\Sigma}_k \coloneqq \boldsymbol{R}_k \boldsymbol{S}_k \boldsymbol{S}_k^\top \boldsymbol{R}_k^\top. \tag{2}$$

In addition to the standard Gaussian parameters such as $\boldsymbol{\mu}_k$, $\boldsymbol{R}_k$, and $\boldsymbol{S}_k$, the Gaussian primitive requires additional learnable parameters for its opacity, $\alpha_k \in [0, 1]$, and feature, $\boldsymbol{f}_k \in \mathbb{R}^d$, which is typically represented by RGB colors or spherical harmonic (SH) coefficients. Thus, each Gaussian primitive is represented as $\gamma_k \coloneqq (\boldsymbol{\mu}_k, \boldsymbol{R}_k, \boldsymbol{S}_k, \alpha_k, \boldsymbol{f}_k)$.

**Differentiable rasterization**  Before rendering with the Gaussian primitives $\Gamma$ on an image space, each 3D Gaussian kernel, $\mathcal{G}_k(\boldsymbol{x}; \boldsymbol{\mu}_k, \boldsymbol{\Sigma}_k)$, is projected onto a 2D image space and forms a 2D Gaussian kernel, $\mathcal{G}_k^\pi(\boldsymbol{r}; \boldsymbol{\mu}_k^\pi, \boldsymbol{\Sigma}_k^\pi)$, where $\pi : \mathbb{R}^3 \to \mathbb{R}^2$ denotes a projection from the world coordinate to an image space. In the projected Gaussian representation, $\boldsymbol{r} \in \mathbb{R}^2$ indicates a pixel location in an image, and the 2D mean $\boldsymbol{\mu}_k^\pi \in \mathbb{R}^2$ and covariance $\boldsymbol{\Sigma}_k^\pi \in \mathbb{R}^2$ are given by

$$\boldsymbol{\mu}_k^\pi \coloneqq \pi(\boldsymbol{\mu}_k) \qquad \text{and} \qquad \boldsymbol{\Sigma}_k^\pi \coloneqq \boldsymbol{J}\boldsymbol{W}\boldsymbol{\Sigma}_k\boldsymbol{W}^\top\boldsymbol{J}^\top, \tag{3}$$

where $\boldsymbol{J}$ denotes the Jacobian of the affine approximation of the projective transformation, and $\boldsymbol{W}$ is the world-to-camera transform matrix. When rendering the primitives in $\Gamma$ to a target camera, they are sorted by their depths with respect to the camera center. The color of a pixel $\boldsymbol{r}$ is then obtained by $\alpha$-blending, which is given by

$$\hat{\mathbf{I}}(\boldsymbol{r}) \coloneqq \sum_{k=1}^{K} \omega_k^\pi(\boldsymbol{r}) c(\boldsymbol{f}_k, \boldsymbol{r}), \tag{4}$$

where $\omega_k^\pi(\boldsymbol{r})$ represents a relative contribution of each Gaussian primitive to pixel $\boldsymbol{r}$ and $c(\boldsymbol{f}_k, \boldsymbol{r})$ is the color of a pixel $\boldsymbol{r}$ measured along the view direction. If a feature vector $\boldsymbol{f}_k$ is based on spherical harmonics coefficients, the color is decoded from $\boldsymbol{f}_k$ using the view direction associated with pixel $\boldsymbol{r}$; otherwise, the feature vector $\boldsymbol{f}_k$ can be identical to the RGB color of the primitive. For more details, please refer to the original Gaussian Splatting paper [23]. Note that, following the $\alpha$-blending procedure in 3DGS [23], $\omega_k^\pi(\boldsymbol{r})$ is given by

$$\omega_k^\pi(\boldsymbol{r}) \coloneqq \alpha_k \mathcal{G}_k^\pi(\boldsymbol{r}; \boldsymbol{\mu}_k^\pi, \boldsymbol{\Sigma}_k^\pi) \prod_{j=1}^{k-1} \left(1 - \alpha_j \mathcal{G}_j^\pi(\boldsymbol{r}; \boldsymbol{\mu}_k^\pi, \boldsymbol{\Sigma}_k^\pi)\right), \tag{5}$$

where $\alpha_k \mathcal{G}_k^\pi(\boldsymbol{r}; \boldsymbol{\mu}_k^\pi, \boldsymbol{\Sigma}_k^\pi)$ is the opacity of the $k^{\text{th}}$ projected primitive at the junction with a ray corresponding to pixel $\boldsymbol{r}$ and $\prod_{j=1}^{k-1}(\cdot)$ is the transmittance at the primitive $\gamma_k$ on pixel $\boldsymbol{r}$, which measure how much light penetrates the preceding primitives along the ray.

## 3.2 Deformation Modeling in 4D Gaussian Splatting

To represent 4D scenes using Gaussian splatting, recent algorithms [68, 18, 61, 31] deform the 3D Gaussian primitives from their canonical states to a target state over time. The transformed position $\boldsymbol{\mu}^t$, rotation $\boldsymbol{R}^t$, and scale $\boldsymbol{S}^t$ at time $t$ are given by

$$(\boldsymbol{\mu}_k^t, \boldsymbol{R}_k^t, \boldsymbol{S}_k^t) = (\boldsymbol{\mu}_k + \phi_\mu(\boldsymbol{\mu}_k, t), \boldsymbol{R}_k + \phi_r(\boldsymbol{R}_k, t), \boldsymbol{S}_k + \phi_s(\boldsymbol{S}_k, t)), \tag{6}$$

where the deformation functions $\phi_\mu(\cdot)$, $\phi_r(\cdot)$, and $\phi_s(\cdot)$ can be various forms, including MLPs, learnable control points [18], Hexplane [5, 61], or polynomial functions. A deformed 3D Gaussian primitive at time $t$ is represented as $\gamma_k(t) := (\boldsymbol{\mu}_k^t, \boldsymbol{R}_k^t, \boldsymbol{S}_k^t, \alpha_k, \boldsymbol{f}_k)$. The projection onto a 2D space follows the same procedure as the static 3D Gaussian splatting, presented in Equation (4). Our approach adopts the Hexplane structure for deformation, similar to [61].

# 4 Uncertanty-Aware 4D Gaussian Splatting

## 4.1 Uncertainty-Aware Regularization

We now discuss the proposed uncertainty-aware regularization technique designed for the balance between reconstruction quality on training images and generalization to unseen views.

**Uncertainty quantification** We first estimate how informative each Gaussian primitive is for reconstruction, based on its visibility from all pixels in training images and its opacity, which is given by

$$C_k = \sum_{\mathbf{I} \in \mathcal{T}} \sum_{\boldsymbol{r}} \omega_k^\pi(\boldsymbol{r}), \tag{7}$$

where $\boldsymbol{r}$ is a pixel in a training image $\mathbf{I} \in \mathcal{T}$, and $\omega_k^\pi(\boldsymbol{r})$ is the contribution of each Gaussian primitive $\gamma_k$ to pixel $\boldsymbol{r}$ during $\alpha$-blending, as described in Equation (5). For the computation of this value, we customize the CUDA kernel to modify the backward process of the 3DGS [23].

The parameters of an informative Gaussian primitive are typically estimated accurately with high confidence. Conversely, a Gaussian primitive that is not properly supported by training images struggles with low accuracy and high uncertainty of its parameter estimation. Based on these observations, the uncertainty of each Gaussian primitive, $U_k$, is defined as

$$U_k = 1 - \texttt{Sigmoid}(C_k; c_0, c_1), \tag{8}$$

where the sigmoid function is used to bound and normalize $C_k$ and $\{c_0, c_1\}$ control the inflection point shift and the slope of the sigmoid function, respectively. Given an arbitrary unseen viewpoint, a 2D uncertainty map $\mathbf{U}$ is constructed using $\alpha$-blending as follows:

$$\mathbf{U}(\boldsymbol{r}) = \sum_{k=1}^K \omega_k^\pi(\boldsymbol{r}) U_k, \tag{9}$$

where $\boldsymbol{r}$ is a pixel in the uncertainty map $\mathbf{U}$. We employ the estimated uncertainty map for the adaptive regularization to unseen views. Specifically, we adopt a diffusion prior as well as a depth smoothing prior and the details of these two priors are discussed next.

**Uncertainty-aware diffusion-based regularization** To render natural-looking images for novel views and times, we incorporate Stable Diffusion [44] into our pipeline. We begin by generating text prompts from the training frames using the vision-language model BLIP [28]. These prompts guide fine-tuning of the diffusion model via DreamBooth [45] with the training images, which aligns the model's understanding to the specific content in the training images as discussed in the image-to-3D reconstruction algorithm [42]. Using this fine-tuned model, we produce a refined image, $\mathbf{I}_{\text{DDIM}}$ from the rendered image $\hat{\mathbf{I}}$ for novel views or times. Specifically, we first encode $\hat{\mathbf{I}}$ into the latent space using the latent diffusion encoder Enc, then perturb it into a noisy latent representation as follows:

$$\mathbf{x}_t = \sqrt{\bar{\alpha}_t}\text{Enc}(\hat{\mathbf{I}}) + \sqrt{1 - \bar{\alpha}_t}\boldsymbol{\epsilon}, \quad \text{where} \quad \boldsymbol{\epsilon} \sim \mathcal{N}(\mathbf{0}, \boldsymbol{I}) \quad \text{and} \quad t \in [0, T], \tag{10}$$

where $\bar{\alpha}_t$ is a scalar value that controls the noise level and $t$ is a diffusion time step. Similar to SDEdit [32], we generate $\mathbf{I}_{\text{DDIM}}$ by performing the DDIM sampling [54] over $k = \lfloor 50 \cdot \frac{t}{T} \rfloor$ steps and running the diffusion decoder Dec as follows:

$$\mathbf{I}_{\text{DDIM}} = \text{Dec}(\text{DDIM}(\mathbf{x}_t, \boldsymbol{v})), \tag{11}$$

where $\boldsymbol{v}$ is the text embedding of the prompt from BLIP. Applying a reconstruction loss between $\hat{\mathbf{I}}$ and $\mathbf{I}_{\text{DDIM}}$ is helpful for generating natural-looking images for unseen views; however, it may compromise reconstruction quality because $\mathbf{I}_{\text{DDIM}}$ sometimes contains misaligned context with the actual 3D scene inherent in the training images. To achieve accurate and realistic reconstruction by balancing the two properties, we propose uncertainty-aware diffusion loss, $\mathcal{L}_{\text{UA-diff}}$. This loss applies the estimated uncertainty to the computation of the reconstruction error between the synthesized image, $\hat{\mathbf{I}}$ and the corresponding DDIM-sampled image, $\mathbf{I}_{\text{DDIM}}$, where uncertain regions are more regularized than low-uncertainty regions—where the training data already provide sufficient reconstruction detail—as follows:

$$\mathcal{L}_{\text{UA-diff}} = \frac{|\mathbf{U} \cdot (\hat{\mathbf{I}} - \mathbf{I}_{\text{DDIM}})|_2}{|\mathbf{U}|_2} + \frac{|\mathbf{U} \cdot (\hat{\mathbf{I}} - \mathbf{I}_{\text{DDIM}})|_1}{|\mathbf{U}|_1}, \tag{12}$$

where $\mathbf{U}$ is the uncertainty map of the unseen view and $\cdot$ denotes element-wise product. Since the DDIM sampling for generating $\mathbf{I}_{\text{DDIM}}$ is time-consuming, performing it every iteration could slow down the training process, which is not desirable. To avoid the computational burden for training, we randomly sample 200 images at the beginning of every 2,000 iterations, refine them by DDIM, and store them in memory. During the next 2,000 iterations, we utilize the stored images to compute the diffusion-based regularization term, $\mathcal{L}_{\text{UA-diff}}$.

**Uncertainty-aware depth smoothing regularization**  We introduce an additional regularization term to encourage smooth depth predictions in uncertain regions. To this end, we first generate a depth map $\mathbf{D}$ for unseen views using an $\alpha$-blending method as follows:

$$\hat{\mathbf{D}}(\boldsymbol{r}) = \sum_{k=1}^{K} \omega_k^{\pi}(\boldsymbol{r}) d_k, \tag{13}$$

where $\boldsymbol{r} \in \mathbb{R}^2$ is a pixel coordinate in the depth map $\hat{\mathbf{D}}$, and $d_k$ denotes the depth at the center of the $k^{\text{th}}$ Gaussian with respect to the camera center. We employ the total variation to regularize the estimated depth map $\hat{\mathbf{D}}$, which promotes smooth depth transitions between neighboring pixels. However, the uniform total variation loss over all pixels produces blur artifacts on accurately predicted regions, resulting in large reconstruction error. To address this drawback, we propose an uncertainty-aware total variation loss, $\mathcal{L}_{\text{UA-TV}}$, which applies stronger smoothing to high-uncertainty regions for noise reduction in the depth map, while leaving low-uncertainty areas unregularized to preserve details. Our uncertainty-aware total variation loss is given by

$$\mathcal{L}_{\text{UA-TV}} = \frac{1}{u_r} \sum_{i,j} \frac{\mathbf{U}_{i,j} + \mathbf{U}_{i+1,j}}{2} \cdot |\hat{\mathbf{D}}_{i,j} - \hat{\mathbf{D}}_{i+1,j}| + \frac{1}{u_c} \sum_{i,j} \frac{\mathbf{U}_{i,j} + \mathbf{U}_{i,j+1}}{2} \cdot |\hat{\mathbf{D}}_{i,j} - \hat{\mathbf{D}}_{i,j+1}|, \tag{14}$$

where $\hat{\mathbf{D}}_{i,j}$ is the estimated depth at pixel $(i, j)$ and $u_r$ and $u_c$ respectively denote the sums of the average uncertainties of vertically and horizontally adjacent pixels, in other words, $u_r = \sum_{i,j} \frac{\mathbf{U}_{i,j} + \mathbf{U}_{i+1,j}}{2}$ and $u_c = \sum_{i,j} \frac{\mathbf{U}_{i,j} + \mathbf{U}_{i,j+1}}{2}$.

## 4.2 Dynamic Region Densification

Existing 4D Gaussian Splattings initialize Gaussian primitives using point clouds obtained from Structure from Motion (SfM) [47]. However, since SfM assumes static scenes, it is fundamentally unable to reconstruct dynamic regions, particularly those involving rapid motion, as shown in Figure 2c. This failure occurs because the algorithm treats dynamic regions as noise, leaving these areas without initialized primitives. Such incomplete initialization disrupts the training process, causing primitives in static regions to be repeatedly cloned and split in an attempt to fill the dynamic areas. This leads to an excessive number of primitives and, in some case, out-of-memory issues.

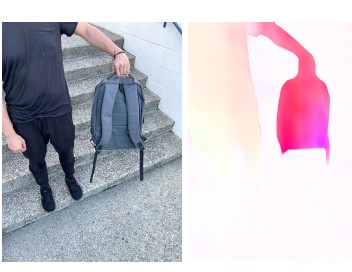 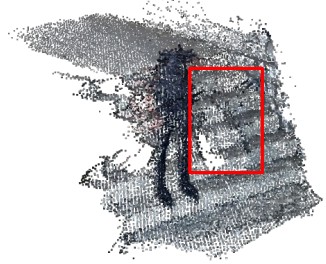 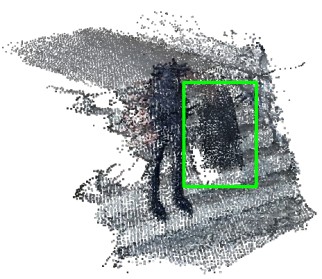

(a) Training image    (b) Scene flow      (c) Initialization from SfM      (d) Dynamic region densification

Figure 2: Visualization of the dynamic region densification on the *Backpack* scene. Since SfM [47] is designed for static scenes, it fails to properly initialize Gaussian primitives in dynamic regions. Our dynamic region densification module initializes additional Gaussian primitives in the identified dynamic regions using scene flow and depth map.

To address this limitation, we propose a dynamic region densification that initializes additional Gaussian primitives $\Gamma' = \{\gamma'_1, ..., \gamma'_K\}$ in dynamic regions. To this end, we first identify dynamic pixels in the training images using scene flows [71] and randomly select a subset of the pixels. For each selected pixel $r$, the corresponding Gaussian primitive $\gamma'_k$ is initialized with position $\boldsymbol{\mu_k}$ and feature vector $\boldsymbol{f_k}$ as:

$$\boldsymbol{\mu_k} = \pi^{-1}\left(\boldsymbol{r}, \mathbf{D}(\boldsymbol{r})\right) \qquad \text{and} \qquad \left[\boldsymbol{f}_k^{(0)}, \boldsymbol{f}_k^{(1)}, \boldsymbol{f}_k^{(2)}\right] = \mathbf{I}(\boldsymbol{r}), \tag{15}$$

where $\mathbf{D}(\boldsymbol{r})$ is the depth value of pixel $r$ estimated from [71], and $\pi^{-1}$ is the inverse projection function that reprojects $r$ into 3D space. The feature vector $\boldsymbol{f_k}$ encodes the information of the $k^{\text{th}}$ Gaussian primitive, where its first three components are set to the RGB colors of pixel $r$ in an image $\mathbf{I}$ and the rest of dimensions, which are optional for the spherical harmonics representation, are padded to zeros. This method provides reasonable placements of Gaussian primitives in dynamic regions as shown in Figure 2d.

### 4.3 Data-Driven Losses

Since the dynamic scene reconstruction from casually recorded monocular video is a highly ill-posed problem, we apply the additional data-driven losses based on depth and flow maps. The depth-driven loss is defined by the difference between the depth maps $\hat{\mathbf{D}}$ and $\mathbf{D}$, which are respectively obtained via the $\alpha$-blending shown in Equation (13) and the algorithm proposed in [71], as shown in the following equation:

$$\mathcal{L}_{\text{depth}} = |\hat{\mathbf{D}} - \mathbf{D}|_1. \tag{16}$$

The flow-driven loss is analogously defined. Given two frames $\Pi$ and $\Pi'$, the flow of a pixel $r$ is estimated using $\alpha$-blending, which is expressed by

$$\hat{\mathbf{F}}^{\Pi \to \Pi'}(\boldsymbol{r}) = \sum_{k=1}^{K} \omega_k^{\pi}(\boldsymbol{r}) F_k^{\Pi \to \Pi'}, \qquad \text{where} \qquad F_k^{\Pi \to \Pi'} = \pi'(\boldsymbol{\mu}_k^{t'}) - \pi(\boldsymbol{\mu}_k^{t}), \tag{17}$$

where $F_k^{\Pi \to \Pi'}$ represents the deformation of the $k^{\text{th}}$ Gaussian primitive from frame $\Pi$ to frame $\Pi'$ in the projected image space, where $(\pi, t)$ and $(\pi', t')$ denote the projection function and timestamp associated with frames $\Pi$ and $\Pi'$, respectively. Similar to $\mathcal{L}_{\text{depth}}$, the data-driven loss for flow is defined by

$$\mathcal{L}_{\text{flow}} = |\hat{\mathbf{F}}^{\Pi \to \Pi'} - \mathbf{F}^{\Pi \to \Pi'}|_1, \tag{18}$$

where $\mathbf{F}^{\Pi \to \Pi'}$ is an optical flow map obtained from RAFT [57].

The final data-driven loss, $\mathcal{L}_{\text{data}}$, is defined as the sum of the two data-driven loss terms, as shown in the following equation:

$$\mathcal{L}_{\text{data}} = \mathcal{L}_{\text{depth}} + \mathcal{L}_{\text{flow}}. \tag{19}$$

## 4.4 Total Loss

To train our uncertainty-award 4D Gaussian splatting models, the total loss function is given by

$$\mathcal{L} = \mathcal{L}_{\text{recon}} + \lambda_{\text{grid}}\mathcal{L}_{\text{grid}} + \lambda_{\text{data}}\mathcal{L}_{\text{data}} + \lambda_{\text{UA-diff}}\mathcal{L}_{\text{UA-diff}} + +\lambda_{\text{UA-TV}}\mathcal{L}_{\text{UA-TV}}, \tag{20}$$

where $\mathcal{L}_{\text{recon}}$ is the standard reconstruction loss based on training images, $\mathcal{L}_{\text{grid}}$ is a loss term associated with the Hexplane-based deformation adopted in our baseline [61], and $\{\lambda_{\text{grid}}, \lambda_{\text{data}}, \lambda_{\text{UA-diff}}, \lambda_{\text{UA-TV}}\}$ are balancing hyperparameters for individual loss terms.

## 5 Experiments

This section compares the proposed method, referred to as UA-4DGS, with existing 4D Gaussian splatting algorithms including D-3DGS [68], Zhan *et al.* [31], and 4DGS [61]. Our method is implemented based on the official code of 4DGS [61] and tested on a single RTX A5000 GPU.

### 5.1 Settings

**Dataset** Our primary goal is to reconstruct dynamic scenes from casually recorded monocular videos, for which we use DyCheck [14] as our main dataset. This dataset consists of monocular videos captured with a single handheld camera, featuring scenes with fast motion to provide a challenging and realistic scenario for dynamic scene reconstruction. The DyCheck dataset includes 14 videos; however, only the half of scenes—*apple*, *block*, *paper-windmill*, *teddy*, *space-out*, *spin*, and *wheel*—are suitable for evaluation due to the availability of held-out views.

**Evaluation protocol** To evaluate novel view rendering quality, we use three metrics: peak signal-to-noise ratio (PSNR), structural similarity (SSIM), and a perceptual metric called learned perceptual image patch similarity (LPIPS). Additionally, since our target dataset provides a co-visibility mask, we also compute masked versions of these metrics, mPSNR, mSSIM, and mLPIPS, focusing on co-visible regions. For masked evaluations, we employ the JAX implementation provided by DyCheck [14].

### 5.2 Experimental Results

Table 1 presents the quantitative comparison of our algorithm against existing methods based on 4D Gaussian Splatting [68, 61, 31] and MLPs [39] on the DyCheck dataset [14]. Our approach, UA-4DGS, surpasses the performance of all other 4D Gaussian Splatting algorithms across all metrics. Figure 3 shows qualitative results on the *space-out*, *paper-windmill*, *teddy*, and *spin* scenes, where UA-4DGS synthesizes more realistic images, clearly outperforming existing 4D Gaussian Splatting algorithms.

Although Gaussian Splatting generally outperforms MLP-based approaches on multi-view or less challenging datasets, our experiments show that they fall behind MLP-based methods in our target setting based on casually recorded videos with a monocular handheld camera as presented in Table 1. This is probably because the methods based on Gaussian Splatting focus more on local optimization with respect to individual Gaussian primitives and, consequently, are prone to overfitting to training images in in-the-wild monocular scenarios.

### 5.3 Analysis

**Generalization to static scenes** To demonstrate the generality of our method in static scene reconstruction, we incorporate the proposed uncertainty-aware regularization to FSGS [72], a few-shot Gaussian Splatting algorithm for static scenes, and refer to this version of our model as UA-FSGS. We test both FSGS and UA-3DGS on the LLFF dataset [33] using three training images with five different runs. Table 2 presents quantitative comparisons where UA-3DGS outperforms existing methods, including both the original results and our reproduced ones of FSGS.

**Ablation study** Table 3 shows the results of the ablation study on each proposed component. Dynamic region densification improves performance compared to the data-driven loss alone, implying that better alignment of primitives with scene geometry enhances the effectiveness of the loss term. Moreover, uncertainty-aware regularization yields further improvements, where $\mathcal{L}_{\text{UA-diff}}$ provides substantial benefits, and adding $\mathcal{L}_{\text{UA-TV}}$ results in additional gains.

Table 1: Quantitative comparisons on a challenging dataset, DyCheck. Our approach shows the best performance among 4D Gaussian Splatting-based methods. However, Gaussian Splatting is generally worse than MLP-based methods in more challenging settings with casually recorded videos using a monocular handheld camera.

| Representation | Method | FPS | mPSNR ↑ | PSNR ↑ | mSSIM ↑ | SSIM ↑ | mLPIPS↓ | LPIPS ↓ |
|---|---|---|---|---|---|---|---|---|
| MLP | T-NeRF [14] | <1 | 16.96 | 16.23 | 0.577 | 0.420 | 0.379 | 0.453 |
| | NSFF [30] | <1 | 16.45 | 15.79 | 0.570 | 0.415 | 0.339 | 0.409 |
| | Nerfies [38] | <1 | 16.81 | 16.43 | 0.569 | 0.417 | 0.332 | 0.399 |
| | HyperNeRF [39] | <1 | 15.46 | 15.20 | 0.551 | 0.399 | 0.396 | 0.464 |
| Gaussian Splatting | D-3DGS [68] | 65 | 12.98 | 12.72 | 0.444 | 0.280 | 0.470 | 0.583 |
| | Zhan *et al.* [31] | **111** | 13.47 | 13.15 | 0.456 | 0.289 | 0.448 | 0.522 |
| | 4DGS [61] | 73 | 14.14 | 13.90 | 0.465 | 0.297 | 0.430 | 0.508 |
| | UA-4DGS (ours) | 75 | **15.25** | **14.89** | **0.488** | **0.325** | **0.390** | **0.476** |

Table 2: Few-shot novel view synthesis results with three views for static scenes, tested on the LLFF [33] dataset. Our method significantly outperforms existing methods across all metrics. FSGS$^\dagger$ and UA-3DGS were tested over five runs, with ($\dagger$) indicating reproduced results. Results for other methods are taken from [72] and [7].

| Representation | Method | FPS | PSNR ↑ | SSIM ↑ | LPIPS ↓ |
|---|---|---|---|---|---|
| MLP | MipNeRF [3] | 0.21 | 16.11 | 0.401 | 0.46 |
| | DietNeRF [20] | 0.14 | 14.94 | 0.370 | 0.496 |
| | RegNeRF [36] | 0.21 | 19.08 | 0.587 | 0.336 |
| | FreeNeRF [67] | 0.21 | 19.63 | 0.612 | 0.308 |
| | SparseNeRF [58] | 0.21 | 19.86 | 0.624 | 0.328 |
| Gaussian Splatting | 3DGS [23] | 385 | 17.83 | 0.582 | 0.321 |
| | Chung *et al.* [7] | – | 17.17 | 0.497 | 0.337 |
| | FSGS [72] | 458 | 20.43 | 0.682 | 0.248 |
| | FSGS$^\dagger$ [72] | **461** | 19.82 | 0.672 | 0.238 |
| | UA-3DGS (Ours) | **461** | **20.89** | **0.716** | **0.217** |

**Impact of uncertainty consideration**   To evaluate the impact of incorporating uncertainty, we test regularization methods without uncertainty consideration, where we refer to this version as $\mathcal{L}_{\text{diff}}$ and $\mathcal{L}_{\text{TV}}$, respectively. Table 4 shows both train and test performance; while $\mathcal{L}_{\text{diff}}$ and $\mathcal{L}_{\text{TV}}$ are still effective for test performance, they exhibit underfitting on training images with lower reconstruction performance. In contrast, by integrating uncertainty through $\mathcal{L}_{\text{UA-diff}}$ and $\mathcal{L}_{\text{UA-TV}}$, we can enhance the balance between training reconstruction quality and test performance in novel view synthesis.

# 6   Conclusions

We proposed a novel training framework for 4D Gaussian Splatting, targeting dynamic scenes captured from casually recorded monocular cameras. Our uncertainty-aware regularizations, which incorporate diffusion and depth-smoothness priors, effectively improve novel view synthesis performance while preserving reconstruction quality on training images. Additionally, we addressed the initialization challenges of Gaussian primitives in fast-moving scenes by introducing dynamic region densification. Our method demonstrated performance gains over baseline approaches, both in dynamic scene reconstructions and few-shot static scene reconstructions. We conducted a detailed analysis through extensive experiments, and we believe this work initiates research on an important, emerging problem in 4D Gaussian Splatting, offering valuable insights to the field.

**Limitations and future work**   Novel view synthesis performance on casually recorded monocular videos still lags behind that on multi-view or simpler datasets, highlighting potential areas for improvement in future research. Currently, our regularization techniques rely on image-level regularization using 2D uncertainty maps; future work could enhance this by incorporating regularization in the Gaussian primitive level [64, 19] to directly leverage each Gaussian primitive's uncertainty. Additionally, our dynamic region densification does not consider temporal consistency for primitive initialization, but this issue may be addressed by integrating long-term tracking algorithms [9].

Table 3: Ablation test results of our training schemes on the *spin* scene in the DyCheck [14] dataset. *Dynamic Dens.* refers to dynamic region densification.

| Method | $\mathcal{L}_{\text{data}}$ | *Dynamic Dens.* | $\mathcal{L}_{\text{UA-diff}}$ | $\mathcal{L}_{\text{UA-TV}}$ | mPSNR | mSSIM | mLPIPS |
|---|---|---|---|---|---|---|---|
| 4DGS [61] | – | – | – | | 15.25 | 0.424 | 0.419 |
| | ✓ | | | | 15.74 | 0.444 | 0.373 |
| | ✓ | ✓ | | | 17.04 | 0.463 | 0.375 |
| Ours | ✓ | ✓ | ✓ | | 17.32 | 0.474 | 0.355 |
| | ✓ | ✓ | ✓ | ✓ | **17.37** | **0.481** | **0.342** |

Table 4: Quantitative comparison of regularization methods with and without uncertainty estimation on the static *room* scene in the LLFF dataset, where FSGS [72] are used as the baseline. Incorporating uncertainty into the regularization improves novel view synthesis by enhancing the balance between reconstruction quality on training images and performance on novel views.

| Method | Uncertainty | Train (Reconstruction) | | | Test (Novel View Synthesis) | | |
|---|---|---|---|---|---|---|---|
| | | PSNR ↑ | SSIM ↑ | LPIPS ↓ | PSNR ↑ | SSIM ↑ | LPIPS ↓ |
| FSGS [72] | – | 42.38 | 0.981 | 0.029 | 20.60 | 0.822 | 0.184 |
| FSGS w/ $\mathcal{L}_{\text{diff}}$ | | 38.40 | 0.986 | 0.035 | 20.40 | 0.811 | 0.198 |
| FSGS w/ $\mathcal{L}_{\text{UA-diff}}$ (Ours) | ✓ | **41.28** | **0.989** | **0.029** | **20.98** | **0.835** | **0.174** |
| FSGS w/ $\mathcal{L}_{\text{TV}}$ | | 38.25 | **0.985** | 0.042 | 20.77 | 0.827 | 0.186 |
| FSGS w/ $\mathcal{L}_{\text{UA-TV}}$ (Ours) | ✓ | **38.26** | **0.985** | **0.038** | **21.08** | **0.831** | **0.184** |

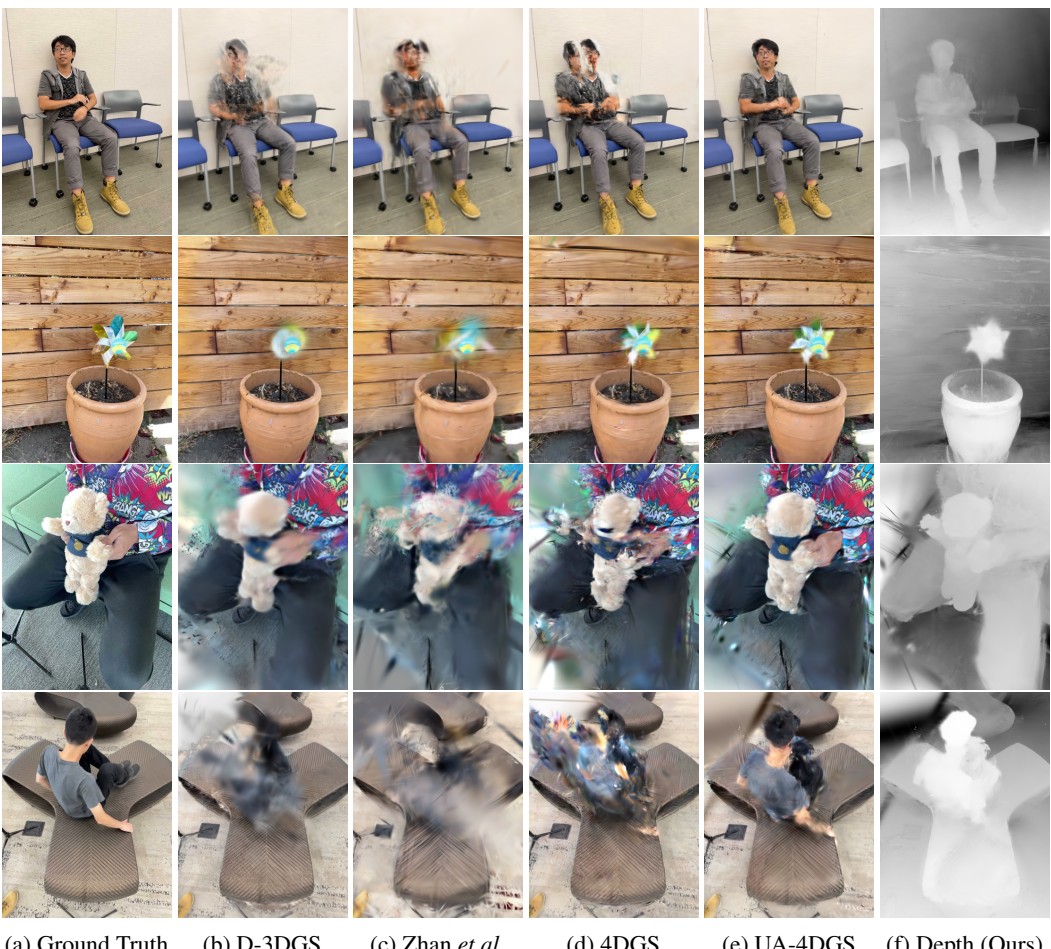

(a) Ground Truth    (b) D-3DGS    (c) Zhan *et al.*    (d) 4DGS    (e) UA-4DGS    (f) Depth (Ours)

Figure 3: Qualitative results on the *space-out*, *paper-windmill*, *teddy*, and *spin* scenes in the DyCheck dataset. UA-4DGS (Ours) shows outstanding quality of rendered images compared to existing methods, including D-3DGS [68], Zhan *et al.* [31], and 4DGS [61]

## Acknowledgments and Disclosure of Funding

This work was partly supported by Samsung Advanced Institute of Technology (SAIT), and by the Institute of Information & Communications Technology Planning & Evaluation (IITP) [No.RS-2022-II220959 (No.2022-0-00959), No.RS-2021-II211343, No.RS-2021-II212068] and the National Research Foundation (NRF) [No.RS-2021-NR056445 (No.2021M3A9E408078222)] funded by the Korea government (MSIT).

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

# A Appendix

## A.1 Per-Scene Breakdown

**DyCheck dataset** Supplementing Table 1 of the main paper, we show the experimental results from individual scenes in terms of mPSNR (PSNR), mSSIM (SSIM), and mLPIPS (LPIPS) in Table 5 As shown in the table, UA-4DGS achieves consistent improvement over its baselines across all metrics.

Table 5: Breakdown results on the DyCheck dataset. The value in $(\cdot)$ represents the non-masked version of the corresponding evaluation measure.

| Method | Apple mPSNR ↑ | mSSIM ↑ | mLPIPS ↓ | Block mPSNR ↑ | mSSIM ↑ | mLPIPS ↓ |
|---|---|---|---|---|---|---|
| D-3DGS [68] | 16.62 (15.69) | **0.711 (0.342)** | 0.391 (0.512) | 12.59 (12.78) | 0.456 (0.287) | 0.553 (0.639) |
| Zhan *et al.* [31] | 15.84 (14.78) | 0.653 (0.275) | 0.436 (0.543) | 13.74 (13.65) | 0.528 (0.385) | 0.482 (0.558) |
| 4DGS [61] | 16.38 (15.35) | 0.691 (0.300) | 0.434 (0.536) | 13.89 (14.11) | 0.548 (0.377) | 0.487 (0.595) |
| UA-4DGS (Ours) | **17.00 (15.66)** | 0.692 (0.313) | **0.375 (0.510)** | **15.84 (15.70)** | **0.588 (0.423)** | **0.430 (0.547)** |

| Method | Paper-windmill mPSNR ↑ | mSSIM ↑ | mLPIPS ↓ | teddy mPSNR ↑ | mSSIM ↑ | mLPIPS ↓ |
|---|---|---|---|---|---|---|
| D-3DGS [68] | 11.399 (11.45) | 0.198 (0.177) | 0.519 (0.656) | 12.42 (12.69) | 0.499 (**0.299**) | 0.491 (0.645) |
| Zhan *et al.* [31] | 13.40 (13.45) | **0.208** (0.187) | 0.417 (0.456) | 11.54 (12.05) | 0.487 (0.259) | 0.519 (0.668) |
| 4DGS [61] | 14.48 (14.52) | **0.208 (0.188)** | 0.375 (0.385) | 12.39 (12.64) | 0.509 (0.275) | 0.486 (0.636) |
| UA-4DGS (Ours) | **14.62 (14.67)** | 0.207 (0.186) | **0.343 (0.351)** | **12.89 (13.02)** | **0.520** (0.291) | **0.476 (0.624)** |

| Method | Space-out mPSNR ↑ | mSSIM ↑ | mLPIPS ↓ | Spin mPSNR ↑ | mSSIM ↑ | mLPIPS ↓ |
|---|---|---|---|---|---|---|
| D-3DGS [68] | 13.10 (14.11) | 0.499 (0.431) | 0.391 (0.455) | 13.30 (12.00) | 0.410 (0.211) | 0.477 (0.681) |
| Zhan *et al.* [31] | 13.94 (14.20) | 0.518 (0.464) | 0.352 (0.424) | 14.04 (13.32) | 0.413 (0.226) | 0.475 (0.552) |
| 4DGS [61] | 14.40 (14.68) | 0.513 (0.447) | **0.365 (0.422)** | 15.25 (14.77) | 0.424 (0.281) | 0.419 (0.511) |
| UA-4DGS (Ours) | **16.24 (16.48)** | **0.546 (0.481)** | 0.368 (0.423) | **17.37 (16.92)** | **0.481 (0.351)** | **0.342 (0.422)** |

| Method | Wheel mPSNR ↑ | mSSIM ↑ | mLPIPS ↓ | Average mPSNR ↑ | mSSIM ↑ | mLPIPS ↓ |
|---|---|---|---|---|---|---|
| D-3DGS [68] | 11.40 (10.31) | 0.334 (0.214) | 0.466 (0.492) | 12.98 (12.72) | 0.444 (0.280) | 0.470 (0.583) |
| Zhan *et al.* [31] | 11.79 (10.56) | **0.385** (0.231) | 0.459 (**0.453**) | 13.47 (13.15) | 0.456 (0.289) | 0.448 (0.522) |
| 4DGS [61] | 12.21 (11.24) | 0.363 (0.211) | 0.445 (0.471) | 14.14 (13.90) | 0.465 (0.297) | 0.430 (0.508) |
| UA-4DGS (Ours) | **12.81 (11.75)** | **0.385 (0.233)** | **0.394** (0.454) | **15.25 (14.89)** | **0.488 (0.325)** | **0.390 (0.476)** |

**LLFF dataset** Supplementing Table 2 of the main paper, we show the experimental results from individual scenes in terms of PSNR, SSIM, and LPIPS in Table 6. As shown in the table, UA-3DGS achieves consistent improvement over its baselines across all metrics.

Table 6: Breakdown results on the LLFF dataset.

| Method | Fern PSNR ↑ | SSIM ↑ | LPIPS ↓ | Flower PSNR ↑ | SSIM ↑ | LPIPS ↓ | Fortress PSNR ↑ | SSIM ↑ | LPIPS ↓ |
|---|---|---|---|---|---|---|---|---|---|
| FSGS [72] | **21.80** | 0.719 | 0.216 | 20.20 | 0.620 | 0.257 | 23.23 | 0.711 | 0.18 |
| UA-3DGS (Ours) | 21.04 | **0.832** | **0.178** | **21.40** | **0.705** | **0.222** | **23.27** | 0.709 | 0.181 |

| Method | Horn PSNR ↑ | SSIM ↑ | LPIPS ↓ | Leaves PSNR ↑ | SSIM ↑ | LPIPS ↓ | Orchids PSNR ↑ | SSIM ↑ | LPIPS ↓ |
|---|---|---|---|---|---|---|---|---|---|
| FSGS [72] | 19.75 | 0.683 | 0.262 | 16.92 | 0.581 | 0.263 | 16.43 | 0.487 | 0.310 |
| UA-3DGS (Ours) | **21.40** | **0.705** | **0.222** | **23.22** | **0.713** | **0.180** | **16.47** | **0.547** | **0.307** |

| Method | Room PSNR ↑ | SSIM ↑ | LPIPS ↓ | Trex PSNR ↑ | SSIM ↑ | LPIPS ↓ | Average PSNR ↑ | SSIM ↑ | LPIPS ↓ |
|---|---|---|---|---|---|---|---|---|---|
| FSGS [72] | 20.60 | 0.822 | 0.184 | **19.65** | **0.755** | **0.225** | 19.82 | 0.672 | 0.238 |
| UA-3DGS (Ours) | **21.21** | **0.833** | **0.178** | 19.11 | 0.685 | 0.264 | **20.89** | **0.716** | **0.217** |

## A.2 Implementation Details

Our method is implemented based on the publicly available official code [1] of 4D Gaussian Splatting (4DGS) [61], using PyTorch [40]. Following our baseline, we utilize the Adam optimizer and set the resolution of the Hexplane grid to $(64, 64, 64, 150)$. For grid smoothness in the Hexplane, we follow the default value of 4DGS. We use the DyCheck dataset [2] as our primary dataset, containing causally captured monocular videos. For diffusion finetuning, we manually select a single prompt from training frames that best represents the overall content of the video. We train our model for 40,000 iterations, where uncertainty-aware regularization is applied starting from iteration 20,000, as the refined images from diffusion model and uncertainty maps become more reliable at this stage. The coefficients $(c_0, c_1)$ for the sigmoid function are set as 0.25 and $20/L$, respectively, where $L$ is the number of training images. We set the balance weights for $\lambda_\text{data}$, $\lambda_\text{UA-diff}$, and $\lambda_\text{UA-TV}$ as 0.5, 0.2, and 0.01, respectively. To measure the quality of generated images, we compute mPSNR, mSSIM, and mLPIPS, leveraging visibility masks provided by the DyCheck.

For experiments on the LLFF dataset, our method is implemented based on the publicly available official code [3] of FSGS [72]. We set the balance weights for $\lambda_\text{UA-diff}$, and $\lambda_\text{UA-TV}$ as 0.1 and 0.001, respectively, for optimal performance, applying the same hyperparameters across all scenes. All experiments are conducted in the Vessl environment [1].

## A.3 Additional Qualitative Results

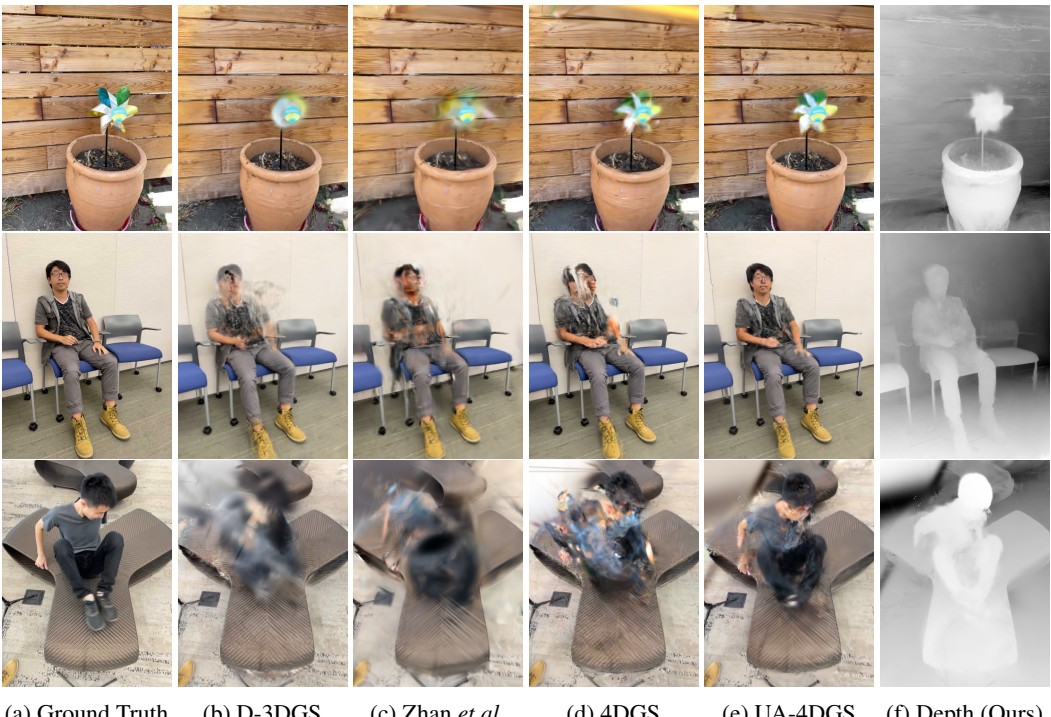

  (a) Ground Truth     (b) D-3DGS     (c) Zhan *et al.*     (d) 4DGS     (e) UA-4DGS     (f) Depth (Ours)

Figure 4: Qualitative comparison between UA-4DGS and other methods tested on the DyCheck dataset. Ours achieves the outstanding quality of rendered images.

---

[1] https://github.com/hustvl/4DGaussians
[2] https://github.com/KAIR-BAIR/dycheck
[3] https://github.com/VITA-Group/FSGS

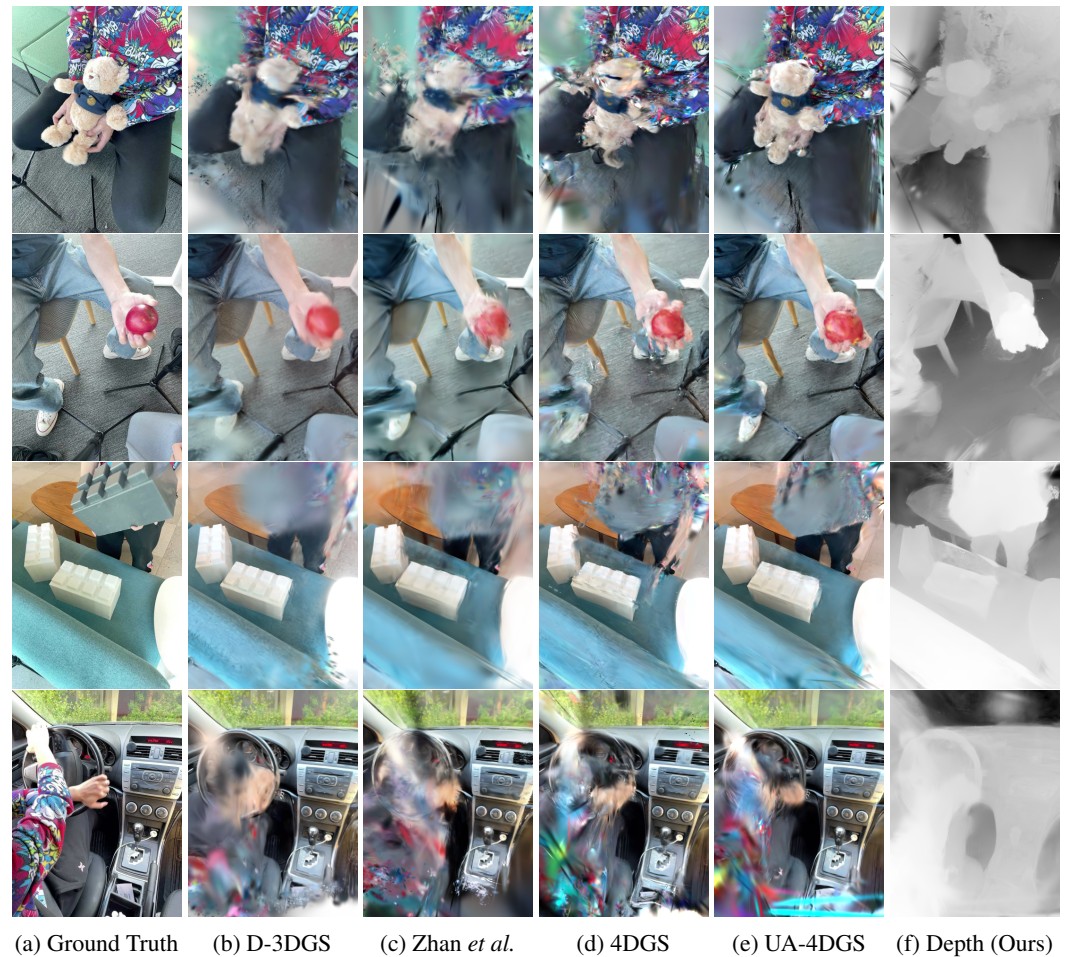

(a) Ground Truth     (b) D-3DGS     (c) Zhan *et al.*     (d) 4DGS     (e) UA-4DGS     (f) Depth (Ours)

Figure 5: Qualitative comparison between UA-4DGS and other methods tested on the DyCheck dataset. Ours achieves the outstanding quality of rendered images.

