# OpenReview forum: "4D Gaussian Splatting in the Wild with Uncertainty-Aware Regularization"
_NeurIPS.cc/2024/Conference — NeurIPS 2024 poster_

### Official Review · Reviewer_fqxz · 2024-07-03

**Soundness:** 2
**Presentation:** 2
**Contribution:** 2
**Rating:** 3
**Confidence:** 5

**Summary:**

The research introduces UA-4DGS, a novel approach designed to reconstruct dynamic scenes from monocular videos. UA-4DGS compensates for information loss due to large motion and self-occlusion by incorporating diffusion priors with pixel-wise uncertainty scores. It proposes an uncertainty-aware diffusion-based regularization, which selectively enhances uncertain regions while maintaining certain ones, preventing inconsistencies with training images. It also identifies and addresses the initialization problem in Gaussian Splatting for dynamic scenes, where static scene-focused Structure from Motion (SfM) techniques fail to initialize dynamic regions properly. They propose dynamic region densification to enhance reconstruction performance and memory efficiency by adding extra Gaussian points to dynamic areas.

**Strengths:**

The paper is presented very clearly, well structured, and it is in general easy to follow and understand.

UA-4DGS's effectiveness is demonstrated using the DyCheck benchmark with complex object motions. Additionally, the research shows that integrating uncertainty considerations into other NeRF regularization techniques can also improve performance.

**Weaknesses:**

1. Writing Quality: The paper contains many typographical and grammatical errors, as well as notation misuse, e.g. L31, 37, 44, 79, and 124. I recommend a thorough review to correct these mistakes.
2. Some assumptions and claims in the paper are questionable and require further validation. For instance, the basic assumption regarding the definition of certainty ("we assume that Gaussians frequently visible in the training images have low uncertainty, while those seen less often due to motion or occlusion have high uncertainty, as they are reconstructed with lower accuracy") and the claim about under-reconstruction in dynamic regions ("under-reconstruction in dynamic regions negatively impacts the training process, resulting in an excessive number of gaussian points and slowing down inference time") need to be substantiated with additional evidence.
3. Experimental results: The experimental results presented in Figure 3 are disappointing. While the proposed UA-4DGS method seems to outperform the baselines, the results are still unsatisfactory, with significant noise and blurriness. Additionally, the comparison is based on only seven images, which is insufficient. I strongly suggest expanding the testing to include more cases to provide a more comprehensive evaluation of the model's performance.

**Questions:**

1. In L171, why use both L1 and L2 losses at the same time?
2. Could you introduce more details about L176 - "Thus, we cache 200 images every 2000 iterations for training efficiency."?

**Limitations:**

The limitations of this paper are primarily found in the method's assumptions and the experimental demonstrations. Please refer to the weaknesses part for more details.

---

> ### Author Rebuttal · Authors · 2024-08-06
>
> Thank you for your constructive comments on improving the writing and analysis of the proposed method. We answered your two concerns on evidence of our assumptions, and lack of experiments as below. We promise to revise our paper by considering your comments including writing quality.
>
> ## W2. Lack of evidence on the assumptions and claims
>
> ### **A. Evidence on the certainty definition**
>
> - We quantify uncertainty based on the visibility in training images. To verify our uncertainty modeling strategy, we measure the estimated uncertainty using AUSE (Area Under the Sparsification Error) [1], which is a common measure for uncertainty quantification [2], where a lower value indicates a high correlation between true error and estimated uncertainty.
>
>
> - For more valid comparisons, we also compare our method with a recent algorithm, FisherRF presented at ECCV24, which suggests a Fisher information-based uncertainty quantification. Note that FisherRF addresses a completely different task, specifically active learning for view selection, whereas our work is the **first to apply uncertainty information for regularization during training**.  Despite this, we demonstrate that our definition is more valid than FisherRF in a sparse setting by showing lower AUSE values (both with MSE and MAE).
>
> |Method 		| AUSE-MSE	 | AUSE-MAE|
> |-----------------|----------------|---------------|
> |Random (upper bound)	| 0.4871	|  0.4878|
> |FisherRF [3]	| 0.4042	|  0.4047 |
> |Visibility (ours)		| 0.3010	|  0.3061|
>
> AUSE-MSE and AUSE-MAE adopt Mean Squared Error (MSE) and Mean Absolute Error (MAE) for the error estimations, respectively. Note that we measure the AUSE in a few-shot setting, which can result in overall higher AUSE values due to the limited data available.
>
> - [1] Estimating and exploiting the aleatoric uncertainty in surface normal estimation. ICCV 2021
> - [2] Bayes’ Rays, CVPR 2024
> - [3] FisherRF: Active View Selection and Uncertainty Quantification for Radiance Fields using Fisher Information, ECCV2024
>
> ### **B. Claims on the importance of dynamic region densification**
>
> - During training, the Gaussian splatting algorithm [4] accumulates gradients of the position values (xy-direction of image space) of each Gaussian. If these gradients exceed a predefined threshold, the Gaussians move, split, or clone to address missing Gaussians in adjacent regions. In dynamic scene reconstruction, no Gaussians initially exist in dynamic regions, causing those in static regions to repeatedly clone and split, leading to a rapid increase in Gaussian numbers.
>
>
> - To verify our claims about over-cloning and over-splitting, we measure the percentage of Gaussians whose gradients exceed the densification threshold, as shown in the table below. Without our dynamic region densification, we observed high gradients in the xy-directions, leading to over-cloning and over-splitting. For better visualization, please see Figure I in the attached PDF, where we illustrate the ratio of Gaussians exceeding the threshold with varying thresholds.
>
> |      Iteration            | w/o dynamic dens.   | w/ dynamic dens. |
> |----------------------|----------------------|--------------------|
> | 600		| 	12.71%		| 	3.38%          |
> | 800		| 	10.09%		| 	3.34%          |
> | 1200		| 	11.65%		| 	3.55%          |
>
> Also, the below table shows the correlation between gaussian numbers and inference speed. It implies that over-densification (over-cloning & over-splitting) reduces inference speed.
> |Gaussian numbers	| Inference speed (FPS) |
> |---------------------|--------------|
> |589102 			| 117.36	|
> |1132420		| 81.89 |
>
>
> [4] 3D Gaussian Splatting for Real-Time Radiance Field Rendering, SIGGRAPH 2023
>
> ## W3. Experimental results
>
>
> We tackle in-the-wild settings where a monocular video contains complex motion. 4DGS approaches deal with monocular video reconstructions, but their target datasets [5,6] are too limited and unrealistic; they only contain small motion or have unrealistic train/test split strategies, making them similar to video interpolation. (The images for training and testing are sampled from a single video stream with a significantly overlapped time interval.)
>
> Again, our target domain is in-the-wild monocular video for real-world scenarios, and it is difficult to find a dataset that satisfies the required conditions. The Dycheck dataset is an example of a realistic video dataset. Please refer to the attached file for more qualitative results.
>
> - [5] HyperNeRF: A Higher-Dimensional Representation for Topologically Varying Neural Radiance Fields. SIGGRAPH 2021
> - [6] D-NeRF: Neural Radiance Fields for Dynamic Scenes. CVPR 2021
>
> ## Q1. L1 and L2 losses
> L1 and L2 losses are often used together in machine learning and optimization tasks to leverage the benefits of both types of loss functions and achieve better regularization. The joint use of two loss terms is helpful for the robustness to outliers (thanks to L1) and the smoothness (thanks to L2). In our problem, the synthesized images should satisfy the properties so the choice of such a loss function is reasonable.
>
> ## Q2 Details of caching methods
> When refining rendered images with diffusion, the DDIM process is quite time-consuming if conducted every iteration. To avoid this, we sample 200 images at the beginning of every 2000 iterations, refine them with DDIM, and cache them. During the 2000 iterations, we utilize these cached images for the UA-Diffusion loss.

---

> > ### Comment · Reviewer_fqxz · 2024-08-10
> > **Thanks for the rebuttal**
> >
> > Dear Authors,
> >
> > I appreciate your efforts in the rebuttal and for providing additional demonstrations. I have carefully reviewed your reply, particularly the examples in the attached PDF. While some cases look very good, most of them appear blurry and noisy in the details. I believe the method still has room for improvement.
> >
> > Thank you.

---

> > > ### Author Response · Authors · 2024-08-11
> > >
> > > We sincerely appreciate the valuable feedback. We have carefully considered the concerns raised in W2 regarding the evidence supporting our assumptions, and we hope that our explanations have provided the necessary clarity.
> > >
> > > Regarding the concerns about qualitative results, as also mentioned by Reviewer 6fVK, we respectfully suggest reviewing our detailed response to Reviewer 6fVK. While we fully acknowledge this concern, we would like to emphasize that our in-the-wild monocular setting dataset is particularly challenging compared to common 4DGS settings. Despite this difficulty, our method still outperforms the existing 4DGS baselines in terms of both qualitative and quantitative results.
> > >
> > > If you have any further questions or would like additional clarification, please do not hesitate to contact us. We would be more than happy to provide additional information or discuss any aspect of our work in greater detail. Your feedback is deeply appreciated, and we remain fully committed to addressing any concerns you may have.

---

> ### Author Response · Authors · 2024-08-13
>
> We sincerely appreciate your valuable feedback. We fully understand and acknowledge your comments, yet we kindly ask that you consider our final, careful clarification regarding our contribution. Your consideration of this would be greatly appreciated.
>
> We have addressed an **emerging and challenging problem of 4D-Gaussian splatting (4DGS)**, especially in in-the-wild settings. Despite the inherent difficulty of the task, by proposing novel training schemes, we have improved both the qualitative and quantitative performance of existing baselines in these challenging settings, as highlighted by Reviewer 6fVK. Additionally, in terms of addressing the emerging and challenging problem of 4DGS, we believe **our paper offers valuable insights and directions that will make a meaningful contribution to the field**.
>
> Regarding the qualitative results, we suggested referring to the response to Reviewer 6fVK in our previous response. As we have not responded directly, we would now like to summarize and emphasize the key points here.
>
> - Our target **in-the-wild dataset is particularly challenging**, as evidenced by existing 4DGS baselines achieving PSNRs below 15 on this dataset, compared to over 25 on other datasets. This difficulty helps explain why the overall qualitative results in this in-the-wild dataset appear lower in quality compared to other commonly used datasets.
> - In the extremely challenging dataset, although some residual blurriness remains, **UA-4DGS consistently outperforms 4DGS baselines and significantly reduces blurriness**, as illustrated in both the main paper and the supplementary pdf file.
> As detailed in our response to Reviewer LdSG, our method also enhances baseline performance on the easier NeRF-DS dataset [1], where overall PSNR is higher and noisy artifacts are almost nonexistent compared to the challenging settings for both the baseline and our method. This suggests that **the blurriness primarily results from the difficulties inherent to the in-the-wild datasets, and our method effectively enhances performance on easier datasets without such issues**.
> - Our training scheme is **general and can be integrated with any 4DGS framework**, potentially enhancing performance through improved Gaussian deformation strategies. We demonstrated this compatibility in our response to Reviewer LdSG.
>
> We appreciate your time and thoughtful feedback, and promise to revise our paper to include the discussions and responses provided during the rebuttal period. If you have any further concerns or questions, please feel free to reply to this message.
>
> [1] NeRF-DS: Neural Radiance Fields for Dynamic Specular Objects. CVPR 2023

---

### Official Review · Reviewer_LdSG · 2024-07-06

**Soundness:** 3
**Presentation:** 4
**Contribution:** 3
**Rating:** 7
**Confidence:** 5

**Summary:**

This paper proposes an uncertainty-aware regularization technique that uses diffusion priors to improve the reconstruction quality of underfitted areas. and a dynamic region densification technique to address the missing initialization problem on dynamic regions.
Experiments verify the proposed techniques.

**Strengths:**

1. The proposed regularization method, which applies uncertainty-aware diffusion priors on unseen views, can reduce inconsistencies with training images.
2. The proposed dynamic region densification can deal with the issue of missing initialization in dynamic regions.
3. The paper is well-written and easy to follow.

**Weaknesses:**

1. The experimental setup is not reasonable. The goal of this paper is to reconstruct dynamic scenes from monocular videos. However, as shown in Sec. 7.2, only one part is relevant to evaluating the proposed method, while the other two involve the generalization of uncertainty-aware regularization. I think the authors should conduct more experiments on the dynamic scene datasets and move the two parts about generalization into the `Appendix`.
2. As shown in L206, the authors claim the proposed method will compare with SC-GS. However, I cannot find such a comparison in Tables and Figures. Moreover, the proposed method should also be compared with Deformable 3D-GS[1].
3. The `Appendix` should be placed after the `References`.

[1] Yang, Z., Gao, X., Zhou, W., Jiao, S., Zhang, Y., Jin, X.: Deformable 3D Gaussians for high-fidelity monocular dynamic scene reconstruction. In CVPR. (2024)

**Questions:**

See weaknesses.
More,
- According to Table 2, for three components of the proposed method, *Dynamic Densification* plays the most important role (+1.3  mPSNR), $L_{\mathrm{data}}$ is the second important (+0.49 mPSNR), and $L_{\mathrm{UA-diff}}$ is the least important. (+0.26 mPSNR). I would like to know the performance of each component when used individually.
- The proposed method seems to be able to change 4D-GS to any other GS-based method. I would like to know the performance of the proposed method when applying it on Deformable 3D-GS or SC-GS.

**Limitations:**

This paper does **not** discuss the limitations and broader impacts.

---

> ### Author Rebuttal · Authors · 2024-08-06
>
> Thank you for complimenting our novel training schemes, uncertainty-aware regularization, and densification technique for monocular video reconstruction, especially having complex object motions. The proposed uncertainty-aware regularization is a generic method, which is effective on both dynamic scene construction and few-shot static scene reconstruction, which is related to weakness 1. We will revise our paper by reflecting on the comments.
>
>
> ## W1. More experiments on dynamic scene reconstruction
> We presented more qualitative results in the attached PDF file in comparison to other methods. The results clearly show that the proposed approach outperforms existing techniques.
>
> In addition, to verify the generality of uncertainty-aware regularization on dynamic scene reconstruction, we evaluate UA-TV loss in a dynamic setting. The table below shows that UA-TV enhances the performance of UA-4DGS for dynamic scene reconstruction.
> This table is an extension of Table 2 in the main paper.
>
> ### Depth regularization on dynamic scene
> |UA-Diff       | UA-TV       | M-PSNR	| M-SSIM	| M-LPIPS        |
> |---------------|----------------|------------------|--------------------|------------------|
> |x             |x               | 17.04	 |0.463		|0.375              |
> |o             |x                |17.30	 |0.474		|0.375              |
> |o              |o                  |17.42	|0.478		|0.374              |
>
>
> ## W2. Missing baselines
> ### Additional baselines on Dycheck datasets
>
> We are sorry about our mistake in L206. We tried to train SC-GS [1] on the Dycheck dataset but failed to optimize the model in most scenes. This is probably because SC-GS targets multi-view synchronized videos in their main paper, which are far from the setting in our mind. Note that, unlike the multi-view setting, the control points in their algorithm are difficult to initialize with only sparse information from in-the-wild monocular video.
>
> In the main paper and the attached file for the rebuttal, we compared our algorithm with three existing techniques [2, 3, 4]. For Deformable 3D-GS [2], we tested it on the in-the-wild Dycheck dataset and visualized rendering results in the attached file. Deformable 3D-GS produces highly blurry images due to the smoothness property of MLP, showing a low mean m-PSNR value of 12.8. Additionally, it sometimes struggles to deform accurately while maintaining a canonical status.
>
> - [1] SC-GS: Sparse-Controlled Gaussian Splatting for Editable Dynamic Scenes, CVPR2024,  code release: 2024.3
> - [2] Deformable 3D Gaussians for High-Fidelity Monocular Dynamic Scene Reconstruction, CVPR2024, code release: 2023.9
> - [3] 4DGS: 4D Gaussian Splatting for Real-Time Dynamic Scene Rendering, CVPR2024, code release: 2023.12
> - [4] Spacetime Gaussian Feature Splatting for Real-Time Dynamic View Synthesis, CVPR2024, code release: 2023.12
>
> ### Limitations of existing baselines and position of our work
> - 4DGS approaches often tackle monocular video reconstructions, but their target datasets are typically unrealistic; they contain limited motion and/or are trivially split into training and testing datasets, making them similar to video interpolation.
> - We first point out that existing 4D-Gaussian splatting paradigms perform poorly on more realistic in-the-wild videos, such as the Dycheck dataset. Our paper partially addressed the existing challenges.
>
>
> ## Q1. Impact of the proposed components
> We agree that the initialization strategy (dynamic densification) has more contribution than UA-Diff, but we emphasize that dynamic densification is also our contribution. Because UA-Diff always has to be applied after dynamic densification to show its effectiveness as shown in the table below, the direct comparison between the two components is fair. Both the modules are the important parts of our algorithm.
>
> |Dynamic dens     | UA-Diff                 | M-PSNR	| M-SSIM	| M-LPIPS |
> |----------------------|-----------------------|---------------|--------------------|------------|
> |x                      |x 		        |15.74	 |0.444	               |0.373       |
> |x                      |o 		        |15.71	 |0.441	               |0.412       |
> |o                        |x 		        |17.04	 |0.463	               |0.375       |
> |o                        |o 		        |17.30	 |0.474                |0.375       |
>
> ## Q2. Plugging into other baselines
> Plugging into other baselines, such as SC-GS or Deformable-3DGS, is a good suggestion, but unfortunately, we need more time for implementation  Currently, we observe that our dynamic densification works well on those baselines. We will discuss more extended experiments during the author-discussion period.
>
> ## L1. Limitations
> Compared to datasets that use multi-view cameras, the in-the-wild setting inherently presents more challenges. Thus, the results is still blurry than other easier datasets. However, our algorithm outperforms the baselines in terms of both qualitative and quantitative results.

---

> ### Comment · Reviewer_LdSG · 2024-08-08
> **Thank you for the rebuttal**
>
> Thank the authors for their detailed rebuttal! It helped with most of my concerns.
>
> However, I still think it is not enough to evaluate the proposed method on only one dataset of dynamic scenes.
>
> There are some datasets recommended:
>
> 1. NeRF-DS dataset used by Deformable 3D-GS
>
> 2. NVIDIA Dynamic dataset used by DynamicNeRF[1]
>
> 3. Unbiased4D Dataset [2]
>
> - [1] Chen Gao, Ayush Saraf, Johannes Kopf, and Jia-Bin Huang. Dynamic view synthesis from dynamic monocular video. In ICCV, 2021
>
> - [2] Johnson, Erik, et al. "Unbiased 4d: Monocular 4d reconstruction with a neural deformation model." In CVPR, 2023.

---

> > ### Author Response · Authors · 2024-08-11
> >
> > We strive to provide comprehensive answers and hope our responses have addressed your queries. Nevertheless, we acknowledge that there might be moments of ambiguity or potential misunderstandings. Please don't hesitate to seek further clarification on any aspect of our work.
> >
> > We're also grateful for the points you raised about compatibility with other 4DGS baselines in Question 2. To demonstrate this, we integrated our proposed methods into the Deformable-3DGS [1]. The table below shows quantitative results, highlighting how our method enhances the performance of another 4DGS baseline.
> >
> > |Method |M-PSNR| M-SSIM| M-LPIPS|
> > |---|-----|-----|----|
> > |Deformable-3DGS [1] |13.75 | 0.398 | 0.495 |
> > |Deformable-3DGS [1] + Ours  | 15.31 | 0.434 | 0.418 |
> >
> > Thank you for suggesting additional monocular video datasets. We tested our algorithm on the plate scene in the NeRF-DS dataset [2], as shown in the table below, further demonstrating the generality of our method. We believe our approach can also be applied to other datasets, including [3,4]. We will incorporate these insights into our revised paper.
> >
> > |Method |PSNR | SSIM	| LPIPS|
> > |---|-----|-----|----|
> > |Deformable-3DGS [1] | 20.48 | 0.812     | 0.222  |
> > |Deformable-3DGS [1] + Ours  |20.74   |0.814 | 0.213 |
> >
> > [1] Deformable 3D Gaussians for High-Fidelity Monocular Dynamic Scene Reconstruction. CVPR 2024
> >
> > [2] NeRF-DS: Neural Radiance Fields for Dynamic Specular Objects. CVPR 2023
> >
> > [3] Dynamic view synthesis from dynamic monocular video. ICCV 2021
> >
> > [4] Unbiased 4d: Monocular 4d reconstruction with a neural deformation model. CVPR 2023

---

> > > ### Comment · Reviewer_LdSG · 2024-08-11
> > > **Thanks for the replay**
> > >
> > > I thank the authors for their efforts. I recommend this paper for acceptance.

---

> > > > ### Author Response · Authors · 2024-08-11
> > > >
> > > > Thank you for your thoughtful feedback on our work and for improving the rating. We promise to revise our paper by considering your comment. If you have any questions related to the work, please don't hesitate to leave comments.

---

### Official Review · Reviewer_6fVK · 2024-07-13

**Soundness:** 3
**Presentation:** 3
**Contribution:** 2
**Rating:** 6
**Confidence:** 4

**Summary:**

This paper tackles the problem of modeling dynamic 3D scenes from monocular videos. To tackle the more challenging dynamic regions in 4D Gaussians, the authors propose to measure the uncertainty and guide those regions with diffusion priors, while keeping certain regions unchanged. In addition, the authors propose to re-initialize the dynamic regions for better performance.

Uncertainty of Gaussian points is measured by their visibility in the training set. Re-initialization of dynamic Gaussian is done by exploiting pre-trained depth and optical flow models.

**Strengths:**

1. Complete ablation studies.
2. Improved quantitative results with respect to the baseline (4DGS).
3. Large quantitative improvements for the sparse view case.

**Weaknesses:**

Major
1. Qualitative results are still of low quality (similar to the previous works).
2. Model training depends on several pre-trained models on different tasks, which increases the complexity of the overall method.

Minor
1. Main performance improvement comes from the initialization strategy
2. Missing definition of co-visibility mask
3. Check the writing continuity of lines 190 and 191

**Questions:**

Where is the "camera information" in line 188, is it from COLMAP?

**Limitations:**

The authors checked the limitation box in the checklist. However, I cannot see the limitations section in the paper nor the appendix.

---

> ### Author Rebuttal · Authors · 2024-08-06
>
> Thank you for acknowledging the strengths of our proposed method, particularly by highlighting the significantly improved performance in both the in-the-wild 4DGS task and the static few-shot task.
>
> ## W1. Low visualization quality
> One of our main contributions addresses the overlooked issue of existing 4D-Gaussian splatting in in-the-wild scenarios. Although there exist 4DGS approaches that deal with monocular video reconstructions, they are limited to tackling videos with limited motion and/or unrealistic train/test split strategies, making the task similar to video interpolation. On the contrary, we target the Dycheck dataset, which is **more realistic and challenging than datasets used in previous research**. Consequently, our qualitative results are of low quality compared to the ones shown in other papers. To demonstrate the superiority of the proposed method, we present more qualitative results together with the outputs from other algorithms, Deformable-3DGS [1], 4DGS, and Spacetime, in the attached PDF file. Note that the additional results can only be evaluated qualitatively because their ground-truths are not available.
>
> Regardless of the quantitative and qualitative improvements, we want to emphasize the value of our work also in the application of uncertainty quantification. Until now, uncertainty quantification in novel view synthesis tasks has been limited to active learning for view selection after training. However, we are the first to leverage uncertainty in the training process, demonstrating its effectiveness in monocular video and few-shot settings.
>
> - [1] Deformable 3D Gaussians for High-Fidelity Monocular Dynamic Scene Reconstruction, CVPR2024
>
>
> ## W2. Complexity of the overall method
>
> Our method relies on several off-the-shelf models such as depth and flow estimators and diffusion models. However, depth and flow estimation models are used only once per example for training. Additionally, since the learnable parameters do not increase, training complexity only increases marginally. Since all the extra models are not required for inference (rendering), our inference cost is identical to the baseline algorithm.
>
>
> ## W3. Impact of proposed components
>
> We agree that the initialization strategy (dynamic densification) has more contribution than UA-Diff, but we emphasize that dynamic densification is also our contribution. Because UA-Diff always has to be applied after dynamic densification to show its effectiveness as shown in the table below, the direct comparison between the two components is fair. Both the modules are the important parts of our algorithm.
>
> |Dynamic dens     | UA-Diff                 | M-PSNR	| M-SSIM	| M-LPIPS |
> |----------------------|-----------------------|---------------|--------------------|------------|
> |x                      |x 		        |15.74	 |0.444	               |0.373       |
> |x                      |o 		        |15.71	 |0.441	               |0.412       |
> |o                        |x 		        |17.04	 |0.463	               |0.375       |
> |o                        |o 		        |17.30	 |0.474                |0.375       |
>
>
> ## W4 & Q1 Clarification
>
> The co-visibility mask, provided by the DyCheck dataset is used for evaluation, assigning a value of 1 to pixels to invisible regions during training. For m-PSNR, m-SSIM, and m-LPIPS, areas where the co-visibility mask is 0 are excluded from the testing process.
> The camera information in lines 190 indicates the camera pose of the training images, which we used as provided by the DyCheck dataset.
>
> ## L1. Limitations
> We apologize for the mistake regarding the checklist and will correct this by adding the limitations. Compared to datasets that use multiview cameras, the in-the-wild monocular video setting inherently presents more challenges. Thus, we agree that the results is still blurry than other easier datasets.

---

> > ### Comment · Reviewer_6fVK · 2024-08-09
> > **Thanks for your reply**
> >
> > Thanks for your answers. I appreciate the additional results, but, as other reviewers pointed out, the qualitative performance is below what NeurIPS expects, making me believe this work still needs more refinement.

---

> ### Author Response · Authors · 2024-08-11
>
> Thank you for your feedback on our results; the quantitative results have significantly improved, but there is still room for enhancement in the qualitative results.
>
> Firstly, we acknowledge this, but it’s important to consider that **our target in-the-wild dataset is particularly challenging**, as evidenced by the fact that **existing 4DGS baselines struggle to achieve a PSNR above 15 in our target dataset, whereas they often exceed 25 on other datasets**. This suggests that it is quite natural for the qualitative results to appear lower in quality compared to those from common 4DGS settings.
>
> Secondly, although UA-4DGS contains some blurriness in the extremely challenging dataset [1] (tackled in our main paper), it **significantly outperforms the 4DGS baselines**, as shown in the attached file. Additionally, in case relatively less challenging NeRF-DS [2] dataset, we observed that our algorithm enhances the baseline with higher fidelity, without blurriness, as demonstrated in our response to Reviewer LdSG. Thus, the blurriness is primarily due to the difficulty of the target in-the-wild datasets, and our method is also capable of enhancing performance in easier datasets.
>
> Finally, our method focuses on the training scheme, not the 4DGS network architecture. Since **the proposed training scheme is indeed general**, it can be seamlessly integrated into existing 4DGS baselines, as demonstrated in our response to Reviewer LdSG. Therefore, we believe our algorithm could achieve even greater performance when combined with future baselines that refine their architecture for better Gaussian deformation strategies. Of course, regardless of the baseline selection, we are committed to continuing our exploration of methods to enhance 4DGS in such a challenging setting.
>
> If you have any further questions or would like additional clarification, please do not hesitate to contact us. We would be glad to provide more information or discuss any aspect of our work in greater detail. Your feedback is deeply appreciated, and we are fully committed to addressing any concerns you may have.
>
> [1] Monocular Dynamic View Synthesis: A Reality Check. NeurIPS 2022
>
> [2] NeRF-DS: Neural Radiance Fields for Dynamic Specular Objects. CVPR 2023

---

> > ### Comment · Reviewer_6fVK · 2024-08-12
> > **Thanks for the clarification**
> >
> > Thanks for your clarifications. I recognize that the proposed method yields better results than the other methods, but it is still far from perceptually pleasant. For this reason, I can only raise my score to weak accept.

---

> > > ### Author Response · Authors · 2024-08-13
> > >
> > > We sincerely appreciate your insightful feedback and the further positive improvement in the rating of our work. We are also grateful for your acknowledgment of the strengths of our method, particularly its advanced qualitative and quantitative performance compared to existing works. Our paper addresses the emerging and challenging problem of 4D Gaussian splatting, especially in in-the-wild settings, and we believe it provides valuable insights and directions that will make a meaningful contribution to the field.
> > >
> > >
> > > Thank you for your time and thoughtful feedback. If you have any further concerns or questions, feel free to reply to this message.

---

### Author Rebuttal · Authors · 2024-08-06

We sincerely appreciate your valuable comments on improving our work. Before answering your questions and concerns, we would like to highlight our contributions.

We present a novel view synthesis algorithm for dynamic scenes captured by a monocular camera. Our target task is fairly new, and we introduce a few interesting ideas such as dynamic densification and visibility-based uncertainty modeling to improve the quality of outputs. Reviewers generally acknowledged our contributions, but expressed concerns about our experiment and presentation. This rebuttal answers the critical questions and concerns raised by the reviewers. We have attached a *one-page PDF* with the characteristics of our model and qualitative results. Please take a look at it and let us know if there are additional concerns or questions.

---

> ### Author Response · Authors · 2024-08-14
>
> We deeply appreciate the insightful reviews and discussions throughout the review period. We will revise our paper to incorporate your valuable feedback. Regarding the reviews, we appreciate that most reviewers gave positive feedback on our work. Below, we summarize the reviews, our responses, and the discussions.
>
>
> In our research, we have addressed an emerging and challenging problem of 4D-Gaussian splatting (4DGS), especially in in-the-wild settings. By proposing a novel training scheme including dynamic region densification and uncertainty-aware regularization techniques, we have improved both the qualitative and quantitative performance of existing baselines, as highlighted by Reviewer 6fVK. As our paper addresses this emerging problem, we believe it offers valuable insights and directions that will make a meaningful contribution to the field.
>
>
>
>
> Upon request by reviewer LdSG, we further investigated the compatibility and generality of our method. To verify compatibility, we integrated our approach with an existing 4DGS baseline, Deformable-3DGS, and demonstrated performance improvements. To demonstrate generality, we further tested our method on an easier dataset compared to our target dataset in the main paper. The results show that our proposed training scheme performs well in both challenging and easier settings, confirming its broad applicability without losing generality.
>
> Reviewer fqxz raised a concern regarding the lack of verification and evidence of our claims. For verification of our uncertainty modeling strategy, we utilized the AUSE (Area Under the Sparsification Error) evaluation protocol to test our modeling strategy. The AUSE results indicate that our uncertainty quantification strategy performs more effectively in sparse settings compared to a recent uncertainty quantification technique. For evidence of dynamic region densification, we first thoroughly explained the principles of densification in Gaussian splatting. We then provided supporting evidence by analyzing xy-directional gradients during training iterations and studying the correlation between inference speed and the number of Gaussians. Our findings reveal that without dynamic region densification, over-densification occurs, which negatively impacts inference speed.
>
> Reviewer 6fVK acknowledged the contributions of our work with a positive rating, yet both reviewers 6fVK and fqxz have raised concerns regarding the qualitative results. As previously mentioned, our paper addresses an emerging and challenging problem, offering valuable insights and directions for the field. Despite the inherent difficulty of the task, our method demonstrates superior performance compared to existing approaches, including Deformable-3DGS, 4DGS, and Spacetime, as illustrated in both the main paper and the supplementary pdf file.
>
> We also clarified some parts needing further explanation, as suggested by all reviewers, including caching strategy, the impact of the proposed components, and limitations.
>
> We believe we have carefully addressed all concerns raised by the reviewers and hope our explanations provide the necessary clarity.
>
> Thank you for your time and feedback.

---

### Decision · Program_Chairs · 2024-09-25

**Decision:**

Accept (poster)

**Comment:**

Reviewers acknowledge the large quantitative improvements (6fVK, fqxz), comprehensive ablation studies (6fVK), the well written paper (LdSG, fqxz) and the ability to integrate diffusion priors for unseen views (LdSG).

The paper received a "weak accept", an "accept" and a "reject". The reviewers voting for "weak accept" and "reject" after the rebuttal have reservations on that there are remaining regions that are not perfectly reconstructed. However, as the authors state, the DyCheck dataset (which has been designed carefully and is accepted in the community), is substantially difficult, and the relevant comparison here is to the baselines. Looking at Figure 3 of the paper, it immediately becomes clear that the method significantly outperforms them. Thus, the AC sees no ground for the reservations and recommends accepting the paper.